# Models Can and Should Embrace the Communicative Nature of Human-Generated Math

**Sasha Boguraev**[†], **Ben Lipkin**[‡], **Leonie Weissweiler**[†], **Kyle Mahowald**[†]

[†]Department of Linguistics, The University of Texas at Austin
[‡]Department of Brain and Cognitive Sciences, Massachusetts Institute of Technology
`{sasha.boguraev, weissweiler, kyle}@utexas.edu`
`lipkinb@mit.edu`

## Abstract

Math is constructed by people for people: just as natural language corpora reflect not just propositions but the communicative goals of language users, the math data that models are trained on reflects not just idealized mathematical entities but rich communicative intentions. While there are important advantages to treating math in a purely symbolic manner, we here hypothesize that there are complementary benefits to treating math as situated linguistic communication and that language models are well suited for this goal, in ways that are not fully appreciated. We illustrate these points with two case studies. First, we ran an experiment in which we found that language models interpret the equals sign in a humanlike way—generating systematically different word problems for the same underlying equation arranged in different ways. Second, we found that language models prefer proofs to be ordered in naturalistic ways, even though other orders would be logically equivalent. We advocate for AI systems that learn from and represent the communicative intentions latent in human-generated math.

> Mathematical propositions are first of all English sentences; not only English sentences, but each mathematical proposition has a resemblance to certain non-mathematical propositions.
>
> —*Ludwig Wittgenstein, Lectures on the Foundations of Mathematics, 1939*

## 1 Introduction

Language Models sometimes rely on heuristics and statistics rather than being perfectly compositional idealized reasoners, especially in domains like math and logic [27, 30, 33, 34, 36, 42, 45, 47]. Whereas language production and comprehension involve some idealized composition using abstract rules [8, 20], in tandem with memorization and pragmatic inference [9, 16], math and logic reflect domains where one might expect an idealized compositional system to be required for obtaining precise solutions. Indeed, whether an expression is written $5 + x = 7$ or $7 - 5 = x$ or "What is 5 less than 7?" or "Seven frogs were sitting on a log. Five left. How many are there now?", there is an underlying computation that can be extracted and performed (namely, the expression $7 - 5$). To properly solve these problems, the thinking goes, systems should abstract away from their situated format into symbolic space.

There is an intuitive, and well-justified, idea that competent human mathematical reasoners employ exactly this kind of abstraction. By contrast, less competent mathematical reasoners (e.g., children struggling to learn math) are often shown to rely on heuristics, schemas, and keywords [6, 10, 24, 38, 48]. For instance, kids might learn that every time they see the phrase "in total" in a word problem, they should add up all the numbers [39]. While the "heuristic" keyword-based direct translation approach may be less cognitively taxing, it is also prone to translation errors [49]. Students who

38th Conference on Neural Information Processing Systems (NeurIPS 2024).

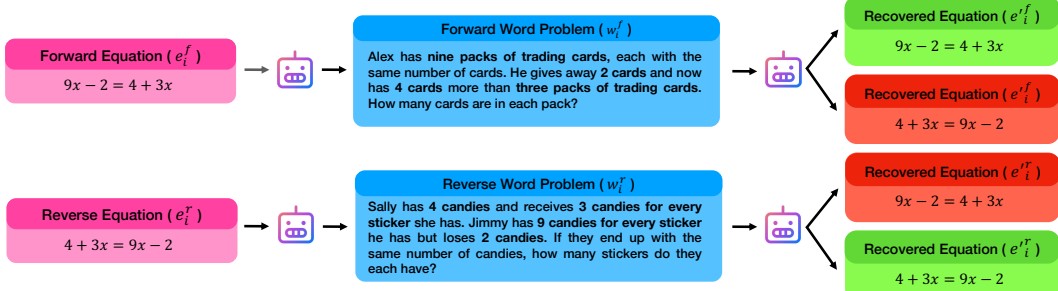

Figure 1: For each pair of equations, we generate corresponding word problems and then try to recover the equations from those problems. The model often recovered the original ordering.

report adopting the more involved strategy of first parsing a math word problem into a structured mental model, then planning computation and finally evaluating the solution in that space, are more successful problem solvers [19].

Taken together, these ideas might make it seem like the goal of AI math models should be to leave the messy domain of language behind and translate expressions into symbolic representations. And, indeed, combining language models with symbolic solvers has proven successful in a variety of math and reasoning domains [4, 14, 18, 32, 46, 53].

Here, we argue that something is lost when disregarding the original context. We introduce the **Communicative Math Hypothesis**: *Math is constructed by people, for people. As such, there are conventions and pragmatics that people bring to the production and comprehension of mathematical expressions—communicative interpretations that go beyond the purely symbolic.* Such traces of information are particularly well suited for study via the tools of linguistics and cognitive science. The choice to write $3x + 9$ instead of $3(x + 3)$ conveys something to the reader, even though they are equivalent. Similarly, the proof of a theorem is not only a formalization that could be computationally verified, but is a communicative act, with intention of being internalized and understood by others.

Drawing on research in math education that we believe is underappreciated in machine learning, we make the case for AI researchers to take the Communicative Math Hypothesis seriously. We present some initial proof-of-concept experiments showing that LLMs pick up on these communicative regularities. We argue that this information should not always be ignored or explained away, but is a crucial component of human mathematics.

## 2 Case Study One: Equations are Asymmetric

Asymmetry in human mathematical interpretation has long been studied in math education. In particular, there is a wealth of literature on the perils of grade-school-aged children's asymmetrical understanding of math – that is, a difficulty in reasoning with a problem such as $\square = 2 + 4$, despite relative comfort with the complementary equation of $2 + 4 = \square$ [2, 37]. But, such sensitivity to asymmetry is not isolated to students. Even expert mathematicians understand math asymmetrically [29], offering different interpretations of equivalent expressions based on what is on the left or right of the equals sign. Here, we present results from a case study demonstrating that LLMs are sensitive to such asymmetries in equations as well and, like humans, do not learn a purely symmetrical interpretation of the equals sign.

**Methods** To test LLMs' sensitivity to symmetry, we conduct an experiment assessing their ability to reconstruct the equations they used to create a specific word problem, as shown in Figure 1. Formally, we perform a three-step experiment. We first generate a set of $n$ paired forward and reverse equations, denoted as $E = \{e_1, e_2, \ldots, e_n\}$, where each paired equation $e_i$ consists of the forward equation $e_i{}^f$ and the reverse equation $e_i{}^r$. Thus, we can express each $e_i$ as $e_i = \{e_i{}^f, e_i{}^r\}$. Next, for each of our $n$ pairs, we pass both equations in $e_i$ to GPT-4o, and prompt it to generate a corresponding pair of word problems, $w_i = \{w_i{}^f, w_i{}^r\}$, that could be solved by $e_i$, with $W = \{w_1 \ldots w_n\}$. We finally ask the LLM to extract the equations $e_i' = \{e_i'^f, e_i'^r\}$ for each $w_i \in W$, with $E' = \{e_1' \ldots e_n'\}$. Our hypothesis is that across all $n$ equations, the LLMs will more often recover $e_i'^f$ from $e_i{}^f$ and $e_i'^r$ from $e_i{}^r$. For details on the equations used, their generation, and model prompting, see Appendix A.

**Results and Discussion**   We measure the recovery rate of an equation's original order and the respective reverse order using GPT-4o across 5 different sets of 200 pairs of randomly generated starting equations. We found that the original equation was recovered on average 52% of the time with a 95% CI of [51%, 54%] across 5 runs. The reverse equation was nearly never recovered: 0.2% of the time, with a 95% CI of [0.0%, 0.4%] – a mere 3 times over the 1000 samples.

These results suggest a difference between the word problems generated from a "forward" equation and word problems generated from a logically equivalent "reverse" equation, and that this difference is itself recoverable by GPT-4o. We posit that this information, which a purely symbolic solver would be agnostic to, is crucial information for systems that aim to use math in collaboration with humans or in human-like ways. These findings are consistent with work showing that premise order matters in LLMs' ability to reason [3, 7, 51], although they frame this order sensitivity as primarily revealing LLMs' brittleness. We interpret these findings (and theirs) as revealing sensitivity to important communicative factors inherent in the data.

## 3   Case Study Two: Mathematical Rules and Proofs Have Orders

Our second case study focuses on mathematical communication of the sort more likely to take place among professional mathematicians: mathematical rules and proofs. Proofs, in particular, are widely used in academic math, as well as related fields, and are duly an area of major focus for AI for math.

Proofs are written to communicate truths that are, in some sense, tautological. Nonetheless, mathematicians have strong expectations and interpretations about the directionality of equation. For instance, there are generalized principles associated with equal signs, like that the right side of the equation expounds upon or explains the left side [29]. Thus, while $a = b$ and $b = a$ are equivalent statements by our agreed-upon set of axioms and inference rules, the choice of one or the other might communicate a different message when used in a proof.

To explore the preferred orderings used by mathematicians in proofs and rules, Mirin and Dawkins [29] utilize a set of *breaching experiments*. Breaching experiments are a class of experiments which try to break rules in an attempt to confirm their existence [41]. In particular, the authors first provided expert mathematicians with a host of formal mathematical equations, such as the distributive rule or an inductive proof. However, these equations were ordered in an unnatural manner – that is, in the case of rules, orders which are not commonly encountered in formal mathematical texts, or in the case of proofs, orders in which steps do not sequentially build from one to the next. The authors measured whether these mathematicians reported any perceived breaches, with any such breaches providing evidence for the existence of the mathematicians' ordering preferences. Our case study into LLM ordering preferences in formal mathematics follows in this vein, measuring LLM surprisals for various natural (extant) and unnatural (unobserved) equation orderings.

**Methods**   Our set of mathematical equations consists of all examples used in the breaching experiments of Mirin and Dawkins [29]. This totals ten different examples, six of which are one line equivalences, expressing common mathematical rules, and the other four of which are a series of equivalences comprising a longer proofs. Each example further contains a brief textual introduction before the series of equivalences. All examples are reported in Appendix B.

We first split each equation into its individual expressions. We then generate every possible ordering of a given equation by permuting the order of these individual expressions. Finally, for each model we calculate the average per-token surprisal for every ordering of expressions in a given equation, conditioned on that equation's textual introductions. Our calculations are performed using the `minicons` package [31], a wrapper around Huggingface's `transformers` package [50].

In this case study, we use the instruction-tuned variants of four models: LLaMa 3.1 8B [12], Mistral 7B v0.3 [23], Mathstral 7B [1], and Qwen2-Math 7B[52]. Two of these models were trained on general corpora (LLaMa and Mistral), the other two fine-tuned on math (Mathstral and Qwen2-Math).

**Equation Variants**   To control for our ten equations potentially being within the evaluated model's training data, we also performed evaluations with three sets of modified, but equivalent, variants of each equation. Our first variants consist of all proofs reworded in a logically equivalent, but expressively different, manner. Our second variants systematically replace all equation variable names, and some rule names, with emojis – maintaining the correctness of equations, but presenting

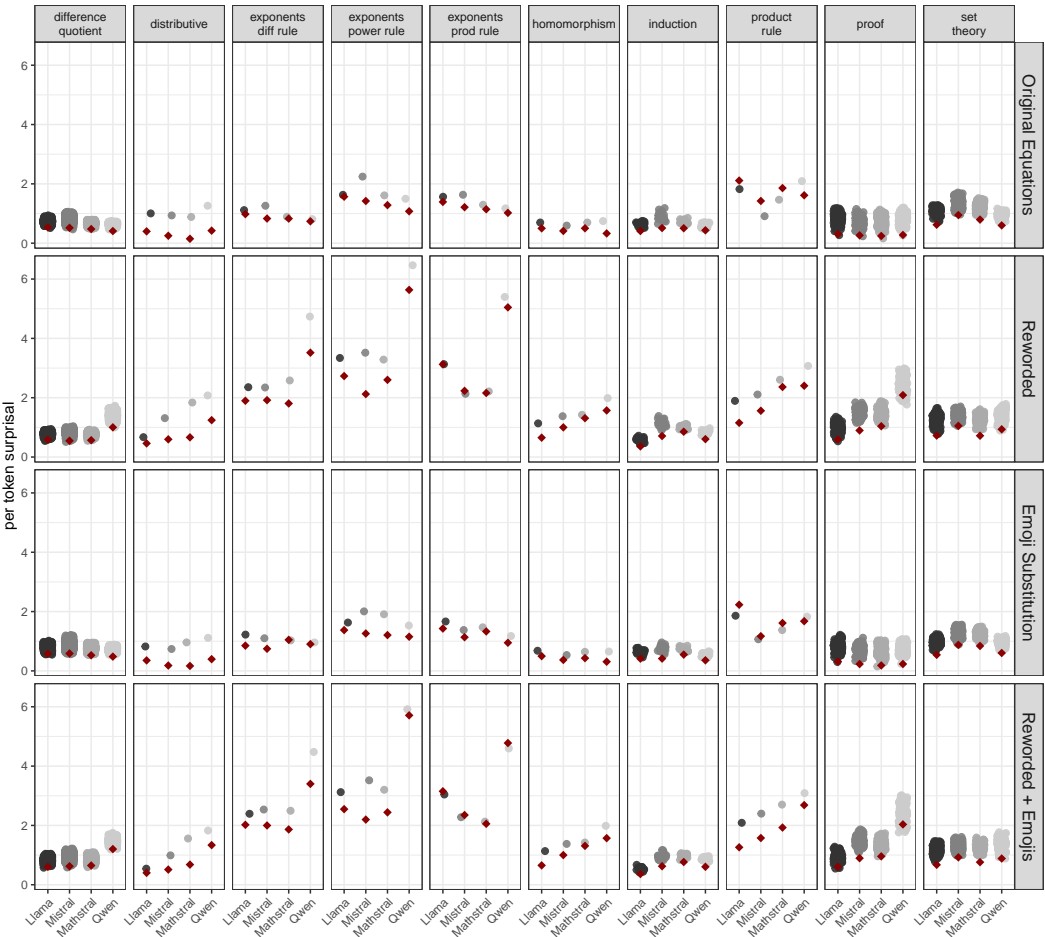

Figure 2: We compare average per-token surprisal for different, logically equivalent orderings of expressions in proofs from Mirin and Dawkins [29] (first row), and corresponding variants (second through fourth row). We find that the original order (◆) has lower per-token surprisals on average (more probable) than equivalent counterfactual orders.

them in a unique, unseen, manner. Our last set of variants combines the two previous variants, substituting emojis into the reworded variants. All variants are reported in Appendix C.

**Results and Discussion** As seen in Figure 2, the evaluated models display clear and consistent preferences for the natural ordering in nine of ten equations. In seven of these, all models display uniform preference for each equation's natural ordering. Of the remaining two equations (DIFFERENCE QUOTIENT and PROOF), only a few nearby orders had a lower surprisal than the natural orders (99.6[th] and 98.3[rd] percentiles, respectively). The only equation for which there is no clear model preference for the natural form is PRODUCT RULE, but this was also a rule noted as unusual by participants in Mirin and Dawkins [29]: mathematicians expressed surprise at seeing $f$ and $g$ instead of $f(x)$ and $g(x)$. When we instead use the latter notation, we see consistent preferences for the natural order. We do not find significant differences between the performances of math-fine-tuned models and more generalized language models across all equations (paired $t = 0.606$, $p = 0.548$).

These results are further consistent across our equation variants (Figure 2). While there is minor variability here (e.g., after rewording proofs, models no longer display clear preferences in EXPONENT PROD RULE but do display clear preferences for natural orderings in PRODUCT RULE) the evaluated LLMs maintain clear and consistent preferences for natural equation orderings even when modified.

These results suggest that LLMs agree with expert mathematicians in their preferences for ordering of proofs and rules, that is in a manner which expresses clear communicative intent. Further, our

work with equation variants suggest that they are aligned due to more than just memorizing training data. This alignment leads to AI systems able to produce math interpretable by those using them, which in comparison to much of the uninterpretable math produced by symbolic solvers and logic programming systems, is a highly desirable quality. As such, while the proofs LLMs produce in their current iteration may not always be correct, any remedies attempting to improve on that correctness should not do so to the detriment of this alignment, if the goal is human use.

## 4   Practical Applications

We focused our experiments on equation asymmetry and proof ordering, showing that LLMs learn extra-symbolic communicative information in both domains. But these principles encompass a much broader class of phenomena. For instance, several patterns identified as reflecting LLMs' brittleness may instead be fruitfully seen as contributing to the communicative interpretation of math.

- Even though they don't matter logically, variable names matter for communicating math (e.g., functions are often $f$ and $g$). This pattern extends to programming as well [21, 28].
- Logically extraneous or pragmatically anomalous information can matter for inferences about how expressions are interpreted [35, 45].
- Notation choice and instruction/prompt phrasing can matter for how problems are solved [17, 22].

Seeing these aspects of LLMs as possible features, and not bugs, could be an important step in developing AI systems that can work with humans. For instance, working mathematicians were long limited to purely symbolic theorem provers. Such systems in isolation neglect the more human aspects of math, ignoring differences in style and comprehensibility. We recommend developing proof assistants that are sensitive to these regularities in human proof-writing and other communicative cues. LLM-based proof systems offer the promise of mathematical assistants that can work *with* people [11], alongside them and not just *for* them as blackbox tools. Below we discuss this idea in two particularly relevant domains.

**Math Education**   Math educators leverage their explicit and implicit understanding of mathematical communicative signaling to enhance teaching. They carefully choose problems presented in manners that probe the intended concepts [26, 44]. They can identify subtle misunderstandings in their students' reasoning just by observing how they discuss mathematical concepts and use them in practice [5, 25, 40, 43]. These are key skills for educators, allowing for more efficient and effective teaching. As we move towards building AI assistants for math education, it is pertinent to develop systems that, like math educators, can both produce and identify these rich communicative signals.

**Math Research**   The furthering of knowledge in any field depends upon the ability to communicate new ideas. If AI math systems are unable to communicate with those using them, we risk merely developing powerful systems that remain limited in their benefits. Instead, we advocate for building systems that produce math in a manner which is communicative and human-like by design, offering promise of furthering our collective mathematical knowledge base. While there is some benefit to systems that can solve problems and prove theorems that humans cannot, what we gain is limited if little information from their methods can be communicated and thusly understood. Of course, AI systems augmented with symbolic solvers do possess the necessary qualities of correctness and robustness which current, non-augmented, LLMs do not. We are not arguing that future AI math systems should lose these qualities, merely that they should also possess communicative sensitivity, something that many current approaches lack. Developing a new generation of hybrid systems, that work through problems via human-interpretable traces, while in parallel formalizing and verifying via symbolic means, appears to be a fruitful path forward.

## 5   Conclusion

While necessarily fuzzier than purely symbolic representations, the communicative principles in human-generated math are not lawless or illogical but can be studied, systematized, and modeled as rational behavior—as they are in linguistics and cognitive science [9, 13, 15]. We join Zhang et al. [54] in their call for a cognitive science perspective on AI and mathematics, centering the role of math as a group activity and communicative endeavor. The math of the people, by the people, for the people, shall not perish from our models.

## Acknowledgments

We would like to thank Paul Dawkins for valuable discussions and insights on mathematical asymmetry and, more generally, the math education literature. We would further like to thank Qing Yao and the computational linguistics research group at UT Austin for their valuable discussions and insights on this work. We would also like to thank Kanishka Misra for assistance with the `minicons` package, and comments on the manuscript. We acknowledge funding from NSF CAREER grant 2339729 (to Kyle Mahowald).

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

## A  Equation Generation and Prompting in Case Study One

### A.1  Equation Generation

For step one of this experiment, we create our equation sets as follows. We first create two independent expressions, each of which consists of two operands, either added or subtracted to each other. One of these operands is a single digit number, with the other being a variable quantity in $x$ with a single digit coefficient. All operands, operations, and choice of which operands are the variable quantity are selected at random. We then form our pair of complimentary equations by placing an equals sign between these two expressions, in both orders. That is, given the two expressions $a$ and $b$, our pair of complimentary equations would be $a = b$ and $b = a$. To illustrate further, a generated set of expressions may include, for example, $2x + 3 = 4 - 5x$ or $8 - 5x = 2 + 3x$, but not $2x = 3$, $4x + 5y = 8x - 2$, or $9x * 2 = x - 4$.

### A.2  Prompting Methods

Our experimental methodology necessitates prompting GPT-4o twice for each equation in our evaluation set: once to create a word problem from a given equation, and once to try and recover an equation given a math word problem. Below, we describe the prompts used for each of these steps.

#### A.2.1  Prompting for Word Problem Creation

For a given equation, EQUATION, we first prime GPT-4o with the following command:

> "You are a helpful middle school math teacher."

We then prompt the model to generate a word problem using the following prompt:

> "Create a grade-school math problem representing the following equation: {EQUATION}. Make sure your problem is clear, concise, represents every term of the equation, and ends in a question mark. Generate just the problem and nothing else."

#### A.2.2  Prompting for Equation Recovery

For a given math word problem, PROBLEM, we first prime GPT-4o with the following command:

> You are a helpful assistant.

We then prompt the model to recover the equation that is represented by PROBLEM with the following prompt:

> "What is the underlying math equation represented by the following situation: {PROBLEM}. Use the letter 'x' for the unknown quantity. Please do not explain, or write any accompanying text, give just a single equation and nothing else."

## B  Equation Set for Case Study Two

We use the following set of equations from Mirin and Dawkins [29] for evaluating model order preferences in formal mathematics. We name each following subsection as they are labeled in Figure 2. Each equation is presented below in its "natural" form. All TeXformatting used to render the following sections, up-to the end of the equations, is included in our experiment.

### B.1  DIFFERENCE QUOTIENT

The **difference quotient** of a function $g$ is defined to be

$$\frac{g(x + h) - g(x)}{(x + h) - x}$$

where $h$ is nonzero. Let $f\colon \mathbb{R} \to \mathbb{R}$ be the function defined by $f(x) = x^2$. The following shows the difference quotient:

$$\begin{aligned}
\frac{f(x+h) - f(x)}{(x+h) - x} &= \frac{f(x+h) - f(x)}{h} \\
&= \frac{(x+h)^2 - x^2}{h} \\
&= \frac{x^2 + 2xh + h^2 - x^2}{h} \\
&= \frac{2xh}{h^2} \\
&= 2x + h
\end{aligned}$$

## B.2  DISTRIBUTIVE

The distributive law tells us that for all numbers $x$, $y$, and $z$,
$$x(y + z) = xy + xz$$

## B.3  EXPONENTS DIFF RULE

Recall the Properties of Exponents:
$$\frac{b^x}{b^y} = b^{x-y}$$

## B.4  EXPONENTS POWER RULE

Recall the Properties of Exponents:
$$(b^x)^y = b^{xy}$$

## B.5  EXPONENTS PROD RULE

Recall the Properties of Exponents:
$$b^x * b^y = b^{x+y}$$

## B.6  HOMOMORPHISM

Let $\langle S, \star \rangle$ and $\langle S', \star' \rangle$ be binary algebraic structures. A **homomorphism from** $\langle S, \star \rangle$ **to** $\langle S', \star' \rangle$ is a function $\phi : S \to S'$ such that for all $x$, $y \in S$,
$$\phi(x \star y) = \phi(x) \star' \phi(y)$$

## B.7  INDUCTION

The following is a portion of a proof by induction that for all natural numbers $k$, $k^3 - k$ is divisible by 6. At this point in the proof, it has been assumed that $n^3 - n$ is divisible by 6, and it is being shown that $(n + 1)^3 - (n + 1)$ is therefore also divisible by 6.

$$\begin{aligned}
(n+1)^3 - (n+1) &= (n^3 + 3n^2 + 3n + 1) - (n + 1) \\
&= (n^3 + 3n^2 + 3n + 1) - (n + 1) \\
&= (n^3 - n) + (3n^2 + 3n) \\
&= (n^3 - n) + 3n(n + 1)
\end{aligned}$$

## B.8  PRODUCT RULE

The *product rule* for derivatives says that if $f$ and $g$ are differentiable functions, then
$$fg' + f'g = (fg)'$$

## B.9 PROOF

**Theorem 1.** *Suppose $\langle S, \star \rangle$ and $\langle S', \star' \rangle$ be binary algebraic structures, and $\phi$ is an isomorphism from $\langle S, \star \rangle$ onto $\langle S', \star' \rangle$. Further suppose that $e$ is a left identity element in $\langle S, \star \rangle$. Then $\phi(e)$ is a left identity element in $\langle S', \star' \rangle$.*

*Proof.* Let $s'$ be an element of $S'$. Since $\phi$ is onto, there exists some $s \in S$ such that $\phi(s) = s'$. Hence

$$s' = \phi(s) = \phi(e \star s) = \phi(e) \star' \phi(s) = \phi(e) \star' s'$$

$\square$

## B.10 SET THEORY

The following is a proof in a set theory textbook that if $a$ is a transitive set, then $\bigcup(a^+) = a$. Note that a transitive set is defined to be a set $a$ such that all members of $a$ are subsets of $a$, and $a^+$ is defined to be $a \cup \{a\}$

*Proof.*

$$\left(\bigcup a^+\right) = \bigcup (a \cup \{a\})$$
$$= \left(\bigcup a\right) \cup \left(\bigcup \{a\}\right)$$
$$= \left(\bigcup a\right) \cup a$$
$$= a$$

$\square$

# C Equation Variants

## C.1 Reworded Variants

### C.1.1 DIFFERENCE QUOTIENT

Let $f \colon \mathbb{R} \to \mathbb{R}$ be the function $f(x) = x^2$. The following shows the difference quotient:

$$\frac{f(x+h) - f(x)}{(x+h) - x} = \frac{f(x+h) - f(x)}{h}$$
$$= \frac{(x+h)^2 - x^2}{h}$$
$$= \frac{x^2 + 2xh + h^2 - x^2}{h}$$
$$= \frac{2xh}{h^2}$$
$$= 2x + h$$

### C.1.2 DISTRIBUTIVE

For all numbers $x$, $y$, and $z$, the distributive law states that

$$x(y + z) = xy + xz$$

### C.1.3 EXPONENTS DIFF RULE

Here are some exponent properties:

$$\frac{b^x}{b^y} = b^{x-y}$$

### C.1.4 EXPONENTS POWER RULE

Here are some exponent properties:

$$(b^x)^y = b^{xy}$$

### C.1.5 EXPONENTS PROD RULE

Here are some exponent properties:

$$b^x * b^y = b^{x+y}$$

### C.1.6 HOMOMORPHISM

If $\langle S, \star \rangle$ and $\langle S', \star' \rangle$ are binary algebraic structures, a **homomorphism from $\langle S, \star \rangle$ to $\langle S', \star' \rangle$** is a function $\phi : S \to S'$ such that $\forall\, x, y \in S$,

$$\phi(x \star y) = \phi(x) \star' \phi(y)$$

### C.1.7 INDUCTION

Inductively prove that $\forall k \in \mathbb{N}$, $k^3 - k$ is divisible by 6. We will show that $(n+1)^3 - (n+1)$ is divisible by 6, with the prior assumption that $n^3 - n$ is divisible by 6.

$$
\begin{aligned}
(n+1)^3 - (n+1) &= (n^3 + 3n^2 + 3n + 1) - (n+1) \\
&= (n^3 - n) + (3n^2 + 3n) \\
&= (n^3 - n) + 3n(n+1)
\end{aligned}
$$

### C.1.8 PRODUCT RULE

If $f$ and $g$ are differentiable functions, then the *product rule* states that

$$(fg)' = fg' + f'g$$

### C.1.9 PROOF

**Theorem 2.** *$\phi$ is an isomorphism from $\langle S, \star \rangle$ onto $\langle S', \star' \rangle$ where $\langle S, \star \rangle$ and $\langle S', \star' \rangle$ are both binary algebraic structures. If $e$ is a left identity element in $\langle S, \star \rangle$, then $\phi(e)$ is a left identity element in $\langle S', \star' \rangle$.*

*Proof.* Let $s' \in S'$. Due to $\phi$ being onto, $\exists s \in S$ such that $\phi(s) = s'$. Hence

$$s' = \phi(s) = \phi(e \star s) = \phi(e) \star' \phi(s) = \phi(e) \star' s'$$

$\square$

### C.1.10 SET THEORY

A transitive set is defined to be a set $a$ such that all members of $a$ are subsets of $a$, and $a^+$ is defined to be $a \cup \{a\}$. We show a proof that if $a$ is a transitive set, then $\bigcup(a^+) = a$.

*Proof.*

$$
\begin{aligned}
\left(\bigcup a^+\right) &= \bigcup (a \cup \{a\}) \\
&= \left(\bigcup a\right) \cup \left(\bigcup \{a\}\right) \\
&= \left(\bigcup a\right) \cup a \\
&= a
\end{aligned}
$$

$\square$

## C.2 Emoji-Substituted Original Proof Variants

### C.2.1 DIFFERENCE QUOTIENT

The 🙂 of a function 😄 is defined to be

$$\frac{😁(😄 + 🙂) - 😁(😄)}{(😄 + 🙂) - 😄}$$

where 🙂 is nonzero. Let 😆: $\mathbb{R} \to \mathbb{R}$ be the function defined by $😆(😄) = 😄^2$. The following shows the 🙂:

$$\frac{😆(😄 + 🙂) - 😆(😄)}{(😄 + 🙂) - 😄} = \frac{😆(😄 + 🙂) - 😆(😄)}{🙂}$$

$$= \frac{(😄 + 🙂)^2 - 😄^2}{🙂}$$

$$= \frac{😄^2 + 2😄🙂 + 🙂^2 - 😄^2}{🙂}$$

$$= \frac{2😄🙂}{🙂^2}$$

$$= 2😄 + 🙂$$

### C.2.2 DISTRIBUTIVE

The distributive law tells us that for all numbers 🙂, 😄, and 😆,

$$🙂(😄 + 😆) = 🙂😄 + 🙂😆$$

### C.2.3 EXPONENTS DIFF RULE

Recall the Properties of Exponents:

$$\frac{🙂^{😄}}{🙂^{😆}} = 🙂^{😄 \text{-} 😆}$$

### C.2.4 EXPONENTS POWER RULE

Recall the Properties of Exponents:

$$(🙂^{😄})^{😆} = 🙂^{😄😆}$$

### C.2.5 EXPONENTS PROD RULE

Recall the Properties of Exponents:

$$🙂^{😄} * 🙂^{😆} = 🙂^{😄 + 😆}$$

### C.2.6 HOMOMORPHISM

Let $\langle 🙂, 😄 \rangle$ and $\langle 🙂', 😄' \rangle$ be binary algebraic structures. A 😆 **from** $\langle 🙂, 😄 \rangle$ **to** $\langle 🙂', 😄' \rangle$ is a function 😆 : 🙂 $\to$ 🙂$'$ such that for all 🙂, 😄 $\in$ 🙂,

$$😆(🙂😄😆) = 😆(🙂)😄'😆(😆)$$

### C.2.7 INDUCTION

The following is a portion of a proof by induction that for all natural numbers 🙂, $😄^3 - 🙂$ is divisible by 6. At this point in the proof, it has been assumed that $😄^3 - 😄$ is divisible by 6, and it is being shown that $(😄 + 1)^3 - (😄 + 1)$ is therefore also divisible by 6.

$$(😄 + 1)^3 - (😄 + 1) = (😄^3 + 3😄^2 + 3😄 + 1) - (😄 + 1)$$

$$= (😄^3 - 😄) + (3😄^2 + 3😄)$$

$$= (😄^3 - 😄) + 3😄(😄 + 1)$$

### C.2.8 PRODUCT RULE

The *product rule* for derivatives says that if 😄 and 😁 are differentiable functions, then

$$(😄😁)' = 😄😁' + 😄'😁$$

### C.2.9 PROOF

**Theorem 3.** *Suppose* $\langle😄, 😁\rangle$ *and* $\langle😄', 😁'\rangle$ *be binary algebraic structures, and* 😂 *is an isomorphism from* $\langle😄, 😁\rangle$ *onto* $\langle😄', 😁'\rangle$. *Further suppose that* 😇 *is a left identity element in* $\langle😄, 😁\rangle$. *Then* 😂(😇) *is a left identity element in* $\langle😄', 😁'\rangle$.

*Proof.* Let 😄' be an element of 😄'. Since 😂 is onto, there exists some 😆 $\in$ 😄 such that 😂(😆) = 😄'. Hence

$$😄' = 😂(😆) = 😂(😇😁😆) = 😂(😇)😄'😂(😆) = 😂(😇)😄'😄'$$

$\square$

### C.2.10 SET THEORY

The following is a proof in a set theory textbook that if 😄 is a transitive set, then $\bigcup(😄^+) = 😄$. Note that a transitive set is defined to be a set 😄 such that all members of 😄 are subsets of 😄, and 😄$^+$ is defined to be 😄 $\cup$ {😄}

*Proof.*

$$\left(\bigcup 😄^+\right) = \bigcup(😄 \cup \{😄\})$$
$$= \left(\bigcup 😄\right) \cup \left(\bigcup\{😄\}\right)$$
$$= \left(\bigcup 😄\right) \cup 😄$$
$$= 😄$$

$\square$

## C.3 Emoji-Substituted Reworded Proof Variants

### C.3.1 DIFFERENCE QUOTIENT

Let 😂: $\mathbb{R} \to \mathbb{R}$ be the function 😂(😁) = 😁$^2$. The following shows the difference quotient:

$$\frac{😂(😁 + 😄) - 😂(😁)}{(😁 + 😄) - 😁} = \frac{😂(😁 + 😄) - 😂(😁)}{😄}$$
$$= \frac{(😁 + 😄)^2 - 😁^2}{😄}$$
$$= \frac{😁^2 + 2😁😄 + 😄^2 - 😁^2}{😄}$$
$$= \frac{2😁😄}{😄^2}$$
$$= 2😁 + 😄$$

### C.3.2 DISTRIBUTIVE

For all numbers 😄, 😁, and 😂, the distributive law states that

$$😄(😁 + 😂) = 😄😁 + 😄😂$$

### C.3.3 EXPONENTS DIFF RULE

Here are some exponent properties:

$$\frac{😄^{😁}}{😄^{😂}} = 😄^{😁 - 😂}$$

### C.3.4 EXPONENTS POWER RULE

Here are some exponent properties:

$$(😀^{😆})^{😆} = 😀^{😆😆}$$

### C.3.5 EXPONENTS PROD RULE

Here are some exponent properties:

$$😀^{😆} * 😀^{😆} = 😀^{😆+😂}$$

### C.3.6 HOMOMORPHISM

If $\langle 😃, 😆 \rangle$ and $\langle 😃', 😆' \rangle$ are binary algebraic structures, a 😆 **from** $\langle 😃, 😆 \rangle$ **to** $\langle 😃', 😆' \rangle$ is a function 😂 : $😃 \to 😃'$ such that $\forall\, 😃, 😆 \in 😃$,

$$😂(😃😆😆) = 😂(😃)😆'😂(😆)$$

### C.3.7 INDUCTION

Inductively prove that $\forall 😃 \in \mathbb{N}$, $😆^3 - 😃$ is divisible by 6. We will show that $(😆 + 1)^3 - (😆 + 1)$ is divisible by 6, with the prior assumption that $😆^3 - 😆$ is divisible by 6.

$$(😆 + 1)^3 - (😆 + 1) = (😆^3 + 3😆^2 + 3😆 + 1) - (😆 + 1)$$
$$= (😆^3 - 😆) + (3😆^2 + 3😆)$$
$$= (😆^3 - 😆) + 3😆(😆 + 1)$$

### C.3.8 PRODUCT RULE

If 😃 and 😃 are differentiable functions, then the *product rule* states that

$$(😃😆)' = 😃😆' + 😃'😆$$

### C.3.9 PROOF

**Theorem 4.** *😂 is an isomorphism from $\langle 😃, 😆 \rangle$ onto $\langle 😃', 😆' \rangle$ where $\langle 😃, 😆 \rangle$ and $\langle 😃', 😆' \rangle$ are both binary algebraic structures. If 😆 is a left identity element in $\langle 😃, 😆 \rangle$, then $😂(😆)$ is a left identity element in $\langle 😃', 😆' \rangle$.*

*Proof.* Let $😆' \in 😃'$. Due to 😂 being onto, $\exists 😆 \in 😃$ such that $😂(😆) = 😆'$. Hence

$$😆' = 😂(😆) = 😂(😆😆😆) = 😂(😆)😆'😂(😆) = 😂(😆)😆'😆'$$

$\square$

### C.3.10 SET THEORY

A transitive set is defined to be a set 😃 such that all members of $a$ are subsets of 😃, and $😃^+$ is defined to be $😃 \cup \{😃\}$. We show a proof that if 😃 is a transitive set, then $\bigcup(😃^+) = 😃$.

*Proof.*

$$\left(\bigcup 😃^+\right) = \bigcup(😃 \cup \{😃\})$$
$$= \left(\bigcup 😃\right) \cup \left(\bigcup\{😃\}\right)$$
$$= \left(\bigcup 😃\right) \cup 😃$$
$$= 😃$$

$\square$

