# OpenReview forum: "Models Can and Should Embrace the Communicative Nature of Human-Generated Math"
_NeurIPS.cc/2024/Workshop/MATH-AI — MATH-AI 24_

### Official Review · Reviewer_eWze · 2024-10-01
**Review for "Models Can and Should Embrace the Communicative Nature of Human-Generated Math"**

**Rating:** 8
**Confidence:** 4

**Review:**

This paper argues that treating mathematical reasoning solely in terms of abstract symbolic manipulation overlooks important communicative and pragmatic aspects inherent in human-generated mathematics. For this central point, as a mathematician myself, I conditionally agree, where the conditions depend on the level of reasoning. In high-level theorem proving or scientific calculations, precision and symbolic rigor may be more important than communicative flexibility. While in more elementary mathematics (especially in education), the communicative nature is crucial.  The authors propose the "Communicative Math Hypothesis," which emphasizes that mathematical language is not purely symbolic but deeply situated within the context of human communication. The paper presents two case studies that explore how large language models (LLMs) like GPT-4 exhibit sensitivity to mathematical expression ordering, suggesting that LLMs can learn and reflect these communicative nuances. Although I believe that this paper is of wide interest to the readership of this workshop, there are some concerns need to be addressed first:

(1)	I think the authors didn’t survey the literature extensively, there are already some literatures on the reversal curse (in similar spirit of LLM’s sensitivity to symmetry discussed in this paper) of LLMs, like The Reversal Curse: LLMs trained on "A is B" fail to learn "B is A" and “Exploring the reversal curse and other deductive logical reasoning in BERT and GPT-based large language models.”
(2)	While the two case studies presented (on equation asymmetry and proof ordering) are interesting, they are somewhat limited in scope. The sample size in the asymmetry experiment is relatively small, and while the results show statistical significance, they lack the robustness that a larger and more diverse dataset could provide. I suggest authors also include different types of equations with multiplications, bracket, division etc. involved. Although I would guess confidently that the same conclusion would still hold, but still I would include them for a more comprehensive set of empirical analysis.
(3)	Following up the previous point, I think authors also need to include more detailed error analysis, especially for the reverse equation. More specifically, I believe that LLMs tend to bias towards certain types of errors and author and this part would be more informative.
(4)	Back to the main point, the paper’s central claim is that models should embrace the communicative nature of math. However, it does not fully address potential counterarguments. As I said, theorem proving or scientific calculations, precision and symbolic rigor may be more important than linguistic communicative flexibility. I suggest authors include a more thorough discussion of when communicative considerations should take precedence would be useful.
(5)	While the authors suggest that LLMs could improve collaboration with humans by better aligning with communicative norms, they do not provide concrete examples of how this would work in practice. Including scenarios where LLMs assist human mathematicians or students in understanding or solving problems could illustrate the real-world impact of their approach. Moreover, I think a user study evaluating how humans interact with communicative versus purely symbolic math AI systems would have strengthened the paper’s claims.

---

### Official Review · Reviewer_t1x1 · 2024-10-05
**Great theoretical idea, further practical usages are highly recommended for the key findings**

**Rating:** 6
**Confidence:** 4

**Review:**

This paper explores a novel perspective on how AI models, particularly large language models (LLMs), should handle mathematical reasoning. The authors propose the "Communicative Math Hypothesis," which emphasizes that mathematics is not just a symbolic manipulation but also a communicative act constructed by humans for humans. The paper demonstrates this hypothesis through two case studies: the asymmetry in equations and the ordering of mathematical proofs. The results suggest that LLMs, like humans, show preferences for certain equation structures and proof orders, which the authors argue should not be disregarded in future AI system designs. The paper skillfully bridges the gap between machine learning, cognitive science, and mathematics. By drawing parallels between linguistic communication and mathematical reasoning, it offers a fresh perspective that could inform the future development of AI systems, especially in fields like education and theorem proving. The interdisciplinary nature of this research enhances its contribution to multiple fields.

However, there are still improvements the authors could make:
While the findings demonstrate that the mathematical expressions generated by the models follow patterns consistent with human conventions, this result is somewhat expected. Transformer-based models, such as those used in this study, are typically trained on large datasets that reflect human-generated language, including mathematical language. Naturally, the sequences generated by these models will align with the common structures found in human-written mathematical expressions. This is largely because the training data itself encodes these conventions, and the models learn to predict the most probable next token based on the patterns they have observed during training. Transformers, particularly when trained with teacher forcing techniques, learn to replicate the input-output mappings found in the data, and thus the generated sequences closely mirror the input format. Although the paper introduces an interesting theoretical perspective, it does not sufficiently address how these findings could be applied in real-world scenarios. For example, it remains unclear how embracing the communicative nature of mathematics in AI models would improve current AI-based mathematical tools, such as those used in education or research. A more detailed discussion on the practical implications and potential use cases of this approach would greatly enhance the relevance and impact of the paper. Without clear pathways to application, the findings risk being seen as more theoretical than actionable.
Besides, in figure 2, it would be great if the authors could explain more about the counterfactual possible orders, inlcuding whether those counterfactual orders are equivalent to the original order or not.
In summary, while the theoretical observations are sound, the findings themselves may not come across as particularly novel or surprising, especially given the underlying mechanics of how LLMs are trained. Furthermore, the paper lacks a clear indication of how these results could be leveraged for future research or practical advancements. Providing more insight into how these findings could influence future studies or be integrated into real-world systems would improve the significance of the contribution.

---

### Official Review · Reviewer_ibJP · 2024-10-05
**This is not a real research paper. The design of experiments is also wrong.**

**Rating:** 1
**Confidence:** 5

**Review:**

This paper explores (1) if large language models can understand asymmetry in equations and (2) if they can detect the ordering of equations.

Even though the authors did some simple experiments, this paper lacks any novelty in methods or results. All they did is putting prompts into large language models and getting the output.

Moreover, experiment 2 is even wrong in its design. The ten examples of equations tested in this paper are all well-known equations and large language models have already seen them millions of times if not billions. If you reorder the equations, no wonder that these models will be surprised.

I also doubt the result of experiment 1. The authors may put the construction step and reconstruction step in one single chat. ChatGPT-4o will then give the same forward/backward equation just based on contexts.

If it was 1998 when Google first launched its search engines, this paper is just like "I found that we can get result A but not B if we search the keyword xxx". This is not a research paper. It can be a blog post though.

---

### Decision · Program_Chairs · 2024-10-08

**Decision:**

Accept

**Comment:**

This paper presents an interesting observation that LLMs can capture more than mathematical reasoning, but also communicative aspects of how humans write mathematics. It should be read more as a position paper, suggesting a space to be explored and motivating it via simple case studies that show the point that LLMs are sensitive to communicative features of language that are formally irrelevant (yet meaningful for humans). I believe it will spark interesting discussions to the workshop. I suggest the authors also discuss potential implications of how the field might concretely respond to the "Communicative Math Hypothesis", and address several of the other points raised by reviewers.